# An Integrated Approach to the Realization of Saudi Arabia's Energy Sustainability

Mohammed Siddig H. Mohammed [1,*], Abdulsalam Alhawsawi [1,2] and Abdelfattah Y. Soliman [1]

1   Department of Nuclear Engineering, Faculty of Engineering, King Abdulaziz University, P.O. Box 80204, Jeddah 21589, Saudi Arabia; amalhawsawi@kau.edu.sa (A.A.); afattah_y@yahoo.com (A.Y.S.)
2   Center for Training & Radiation Prevention, King Abdulaziz University, P.O. Box 80204, Jeddah 21589, Saudi Arabia
*   Correspondence: mshmohamad@kau.edu.sa

**Abstract:** As system thinking is a recognized approach to the comprehension and realization of energy sustainability, this paper applies a holistic representation to the World Energy Trilemma Index (WETI) key indicators using Bayesian Belief Networks (BBN) to illuminate the probabilistic information of their influences in Saudi Arabia's context. The reached realization is suggested to inform the policies to improve energy sustainability, and thus the country's rank in the WETI. The analysis used two groups of learning cases, one used the energy statistics of the period from 1995 to 2019 to show the outlook of the Business as Usual path, and the other addressed the projected data for the period from 2018 to 2037 to investigate the expected impact of the new policies. For both BAU and new policies, the BBN calculated the improvement, stability, and declining beliefs. The most influential factors on energy sustainability performance were the electricity generation mix, $CO_2$ emissions, energy intensity, and energy storage. Moreover, the interlinkage between the influential indicators and their causes was estimated in the new policies model. A back-casting analysis was carried out to show the changes required to drive the improvement belief to 100%. The compiled BBN can be used to support structuring policymaking and analyzing the projections' outcomes by investigating different scenarios for improvement probabilities of energy sustainability.

**Keywords:** Saudi Arabia; energy sustainability; world energy trilemma index; Bayesian Belief Network

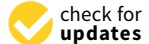



## 1. Introduction

The United Nations' (UN) 2030 agenda for Sustainable Development was announced in 2015 under the main title and objective of "Transforming Our World," and indeed, the world is witnessing transformations that are implemented and measured by the 17 Sustainable Development Goals (SDGs) and their indicators [1].

The SDG seven aims to ensure access to affordable, reliable, sustainable, and modern energy for all the world's population. The interaction of SDG 7 targets with the targets of SDG 1: No poverty, SDG 2: Zero hunger, SDG 3: Good health and wellbeing, SDG 6: Clean water and sanitation, SDG 8: Decent work and economic growth, and SDG 13: Climate actions have been analyzed and the mutual impacts identified to be reinforcing, enabling, or constraining. In addition, the mutual influence between SDG 7 and the other 16 SDGs has been presented [2], which demonstrated that energy sustainability is crucial for sustainable development.

The discussion and analysis of energy sustainability have been approached in the literature by several methods. A procedure has been proposed to evaluate the Sustainable Useful Index of energy-producing processes. The index assesses the ability to maintain the viability and usefulness of energy sources considering the produced, spent, avoided, and direct energy. The definition does not satisfy the broad concept of energy sustainability [3]. Within the same perspective, energy sustainability analysis has been performed at short-

term and long-term levels and applied to a case study on the production of distributed $H_2O$ [4].

Energy sustainability analysis for developing countries in the light of SDGs has been attempted. However, the authors limited the discussion on the role of a hybrid power system in improving energy sustainability in urban areas making a case study based on data from an Iranian city [5].

The authors of [6] have studied non-renewable energy and renewable energy efficiency for simultaneous achievement of economic growth and environmental sustainability in the Middle East and North Africa. The scope of the study has not included some important aspects of energy sustainability, such as the social dimension (energy equity and affordability).

Grigoroudis et al. presented a definition and mathematical model based on various indicators that cover the adequacy, reliability, affordability, social, and environmental requirements. The model rates energy sustainability on a 0 to 1 scale using Fuzzy Logic reasoning. The methodology is more integrated than the previous studies, however, one shortcoming is the potential subjectivity associated with assessment by Fuzzy Evaluation [7].

One of the comprehensive and informative methodologies on the performance of energy sustainability is World's Energy Trilemma Index (WETI) [8], published since 2010 by The World Energy Council, the UN's accredited energy body. Consistent with the World Energy Council definition of energy sustainability, the WETI ranks the countries energy performance on three dimensions: Energy security, energy equity, and environmental sustainability. The index helps to assess the effectiveness of energy policies for enabling balanced transition management, perform a comparative analysis using the experiences of countries with relevant socioeconomics and energy infrastructure, and eventually inform the policies on the required adjustments.

The WETI has been investigated using several methods to assess its reliability. The research in [9] has praised the value of the WETI in guiding countries to address the energy trilemma. Nonetheless, it has argued that the preferences among the trilemma can change from country to country, which requires weighing the trilemma dimensions adaptably. The research suggested the use of interval decision making and stochastic multicriteria acceptability analysis to measure countries' energy performance, and developed an alternative ranking scheme. In another contribution to addressing the preference variation of the trilemma dimensions, the interval decision matrix and the principal component analysis were used to evaluate the top ten performers in the 2015 version of WETI and produce a comparison rank that debated the weights assigned to the WETI indicators [10]. Principal Component Analysis has been applied to assess the methodology of the WETI using Pearson Correlation test, Kaiser–Meyer–Olkin Measure of Sampling Adequacy and Bartlett's Test of Sphericity. The conclusion was made from the results of Cronbach's Alpha test, and the authors deemed the WETI ranking unreliable [11].

Nevertheless, the methodology of WETI has been revised over the years since its first release, the most recent revisions were in the 2019 version that included the data sources, indicators, weighting, and indexation [8] (p. 42) and further refinements in 2020 report [12] (p. 63).

The ranking is based on grading the performance of each of the dimensions from A to D. Although Saudi Arabia is the world's leading oil producer and is among the top 20 countries in terms of Gross Domestic Product (GDP) [13], it was ranked 78 out of 127 in the 2019 WETI, mainly because of the poor performance in the Environmental Sustainability dimension scoring 35/100 (D) and the average performance in the Energy Security 55/100 (C), while scoring 98/100 (A) in the Energy Equity.

Saudi Arabia's ranking in 2019 retreated from positions 47, 53, and 47 in the years 2016, 2017, and 2018, respectively. In the years from 2016 to 2018, Saudi Arabia kept a consistent grade of BAD. Therefore, the comparative ranking fluctuation could be attributed to the performances of other countries as their ranks were rolling up and down. The downgrade in 2019 ranking is mainly due to the decline in the energy security dimension that has

consistently been graded with B due to lack of energy diversity in the past three years, to a C grade in 2019, which again can be attributed to the outpace of other countries besides the geopolitical tensions in the region. The strength of Saudi Arabia in the energy trilemma is in the energy equity dimension due to the availability and affordability of fuel and electricity.

The 2020 WETI was released recently [12]. Saudi Arabia is ranked 55 out of 108 countries, however, the progress from 78 in 2019 does not indicate a corresponding improvement in energy sustainability performance since the grade is still BAD. The quasi progress could be because fewer countries were included in 2020 and also because of the performance swings of the other countries.

In 2018, Saudi Arabia's total primary energy supply was 133,291 ktoe of oil and 78,009 ktoe of natural gas, 221,836 GWh of the electricity was generated from natural gas, 125,860 GWh from oil, and 155 GWh from solar Photovoltaic (Solar PV) [14]. The access to electricity covers 100% of the population [15], and the fuel and electricity prices, although they have recently been witnessing subsidies reforms, are still affordable. The presented figures explain Saudi's Energy Trilemma Index high score in energy equity and the low score of environmental sustainability. Moreover, the statistics show the reliance of the Saudi economy and the energy sector on oil and gas consumption and exports, which impacts the energy security score.

However, in recent years, and simultaneously with the efforts of the world to undergo an unprecedented transition to sustainable development, Saudi Arabia has announced an ambitious transformation plan known as Vision 2030 [16] that was built around three pillars: A vibrant society, a thriving economy, and an ambitious nation. One of the objectives of the vibrant society pillar is the maintenance of environmental sustainability, with one of its measures being the reduction of air pollution. The thriving economy pillar mainly aims at economic diversification, and it includes the objective of introducing renewables to the country's energy mix, increasing the production of natural gas, and controlling energy consumption by introducing plans for fuel-targeted subsidies. The objectives mentioned above support the implementation of programs that can lead to enhancing energy security and environmental sustainability.

There are two main trends in the literature about the energy sector in Saudi Arabia, one is about alternative energy sources, mainly renewables, and the other is on energy economics. The status and potential of renewable energy resources in Saudi Arabia have been reviewed, and the possible roles of renewable energy in developing policies for secured and cost-effective energy have been examined [17]. Renewable energy solutions for the challenge of increasing oil consumption in Saudi Arabia have been discussed as well as the outlook of energy cost and clean environment [18]. The human resources requirements to meet the future of renewables in Saudi Arabia have been presented [19].

Regarding energy economics, different policy scenarios to decouple the reduction in fuel consumption and energy cost increase or optimizing the prices of industrial fuels and household electricity [20–22] to seek a more efficient energy system have been discussed.

The presented literature gives useful insights and solutions, however, they considered siloed elements of the energy system, which have not addressed the holism of energy sustainability.

This work proposes to apply the holistic approach of the system thinking [23] utilizing Bayesian Belief Network (BBN) to examine the influences of the indicators underpinning the implementation of energy security, energy equity, and environmental sustainability in Saudi Arabia's context. The reached comprehension uncovers the probabilities of the impact and mutual interactions between the indicators and the likelihood of changes. The proposed method can support decision-making in energy policy prioritization, schedule, or amendments that can result in the improvement of Saudi's energy sustainability and WETI rank.

## 2. Materials and Methods

BBN represents the probabilistic relationships between a set of variables in a Directed Acyclic Graph (DAG). The DAG is composed of nodes to denote the variables and links (arrows) to represent the causal connection between the variables. The relationship between the causes and effects is described by Conditional Probability Tables (CPT) to identify the belief that the effect variable will be in a specific state given the state of the cause variable.

If a state of a variable is changed, the change is transmitted through the links, and the network is solved using Bayes' theorem.

$$P(A|B) = \frac{P(B|A)\ P(A)}{P(B)} \tag{1}$$

where P(A) is the prior distribution of variable A, P(A | B) is the posterior distribution (the probability of A given new data B), and P(B | A) the likelihood function [24] (p. 6).

Introductions and a detailed formal definition of BBN are given in [25–27].

In energy systems and energy policy, the BBN has been for providing a tool for policymaking in the renewable energy sector [28], decision-making in clean energy investment [29], assessment of power systems [30], and the integration of renewables into the grids [31].

BBN is used in this paper to examine the influences of some WETI indicators on energy sustainability in Saudi Arabia.

The calculation of the WETI is based on 32 indicators, however, twelve key metrics are used in the countries' profiles to exhibit the performance. This paper considers nine key metrics shown in Figure 1, generated using Vensim system dynamics simulation software. The remaining three key metrics of the fourth dimension, the country context, are beyond the current scope.

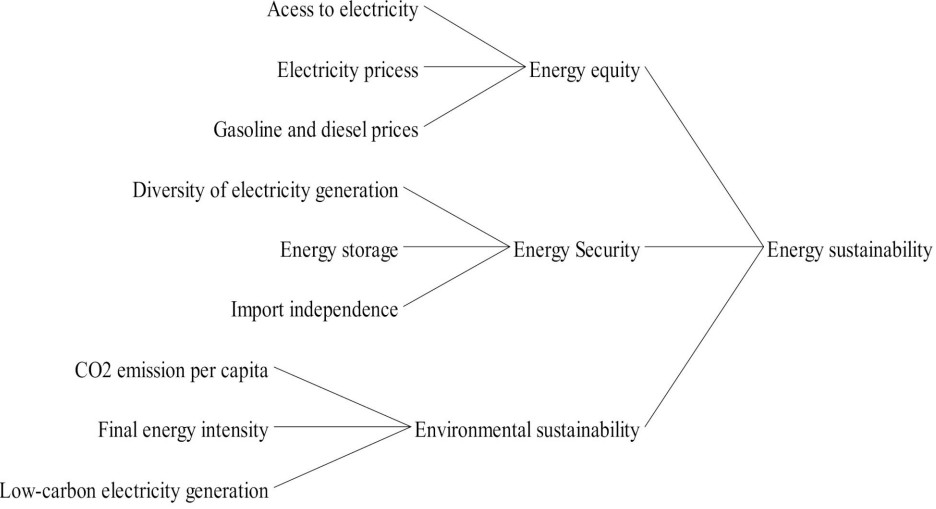

**Figure 1.** Causes tree energy sustainability using the selected key metrics.

Cases of examples or experiences data are provided to train the BBN to capture the believed states in different scenarios. The datasets measuring the indicators between the years 1990 and 2019 were obtained from various sources cited in Table 1, used to train the constructed BBN. Table 1 contains 25 cases, with each row being a case.

**Table 1.** Energy statistics for the period from 1990 to 2019. The asterisks mean missing data.

| | Energy Imports % of Energy Use [32] | Oil Refinery Capacities (Million Barrels/Day) [33] | Share of Renewables in the Electricity Mix % [34] | Energy Intensity (kW-h/2011$) [35] | CO$_2$ (Tons/Capita [36] | Elect. Prices SR/kWh [37] | Gasoline Prices SR/Liter [38–40] | Access to Elect. (% of Population) [15] |
|---|---|---|---|---|---|---|---|---|
| 1990 | −535.188 | 1.86 | * | 3.212282 | 11.42588 | 0.07 | * | 100 |
| 1991 | −586.682 | 1.645 | * | 3.549141 | 15.94175 | 0.05 | * | 100 |
| 1992 | −520.327 | 1.66 | * | 2.773358 | 16.49478 | 0.05 | * | 100 |
| 1993 | −490.736 | 1.67 | * | 3.076266 | 17.63903 | 0.05 | * | 100 |
| 1994 | −449.632 | 1.683 | * | 3.801903 | 16.88071 | 0.05 | * | 100 |
| 1995 | −449.353 | 1.692 | * | 3.717125 | 12.59267 | 0.05 | 0.16 | 100 |
| 1996 | −418.536 | 1.699 | * | 3.617888 | 13.56865 | 0.05 | * | 100 |
| 1997 | −438.437 | 1.704 | * | 3.640062 | 11.11945 | 0.05 | * | 100 |
| 1998 | −420.669 | 1.762 | * | 4.066808 | 10.47502 | 0.05 | 0.16 | 100 |
| 1999 | −372.668 | 1.808 | * | 3.592834 | 11.18999 | 0.05 | * | 100 |
| 2000 | −386.248 | 1.798 | * | 3.155803 | 14.34164 | 0.05 | 0.24 | 100 |
| 2001 | −361.93 | 1.805 | * | 3.511994 | 13.98764 | 0.05 | 0.24 | 100 |
| 2002 | −290.774 | 1.809 | * | 3.292017 | 14.93586 | 0.05 | 0.24 | 100 |
| 2003 | −349.868 | 2.049 | * | 3.255635 | 14.53989 | 0.05 | 0.24 | 100 |
| 2004 | −352.903 | 2.074 | * | 2.945454 | 17.05746 | 0.05 | 0.24 | 100 |
| 2005 | −365.873 | 2.102 | * | 2.584772 | 16.62125 | 0.05 | 0.24 | 100 |
| 2006 | −316.28 | 2.102 | * | 2.372616 | 17.60497 | 0.05 | 0.16 | 100 |
| 2007 | −289.452 | 2.102 | * | 2.291149 | 15.34697 | 0.05 | 0.16 | 100 |
| 2008 | −266.008 | 2.102 | 0.000277 | 2.04159 | 16.69991 | 0.05 | 0.16 | 100 |
| 2009 | −211.609 | 2.109 | 0.000262 | 2.424205 | 17.49302 | 0.05 | 0.16 | 100 |
| 2010 | −186.51 | 2.109 | 0.001631 | 2.272282 | 18.87995 | 0.05 | 0.16 | 100 |
| 2011 | −232.737 | 2.107 | 0.001907 | 1.866232 | 17.60523 | 0.05 | 0.16 | 100 |
| 2012 | −211.956 | 2.107 | 0.008289 | 1.869318 | 19.31661 | 0.05 | 0.16 | 100 |
| 2013 | −219.74 | 2.507 | 0.013063 | 1.903283 | 17.99534 | 0.05 | 0.16 | 100 |
| 2014 | −191.524 | 2.899 | 0.012645 | 2.054486 | 19.46813 | 0.05 | 0.16 | 100 |
| 2015 | * | 2.899 | 0.035919 | 2.034974 | 19.5753 | 0.05 | * | 100 |
| 2016 | * | 2.901 | 0.034878 | 2.032694 | 19.46668 | 0.05 | 0.2 | 99.9 |
| 2017 | * | 2.826 | 0.037127 | * | 19.12645 | 0.05 | * | 99.93 |
| 2018 | * | 2.835 | 0.040355 | * | 18.43629 | 0.18 | 0.36 | 100 |
| 2019 | * | 2.835 | 0.198166 | * | | 0.18 | 0.36 | 100 |

The oil refinery capacities were used to indicate the energy storage capacity, and the share of renewables in the electricity mix was used as an input for two indicators: Diversity of electricity generation and low-carbon electricity generation.

The selected metrics were represented using BBN. The states of the variables in the BBN were drawn from the data in Table 1. The states were described as declining, stable, or improving according to the comparison of the measurement of the specified year to the average of the preceding five years. If the change percent is zero, the state is named stable. Positive and negative percentages are named improving or declining depending on the specific indicator, for example, a negative change in CO$_2$ emission is an improvement. The states of the indicators are given in Table 2.

However, the changes in the electricity generation sources were treated differently because the growth in the renewables shares in the electricity mix was negligible from 2008 to 2014, so they were not considered improvements. Then, from 2015 to 2019, the states were based on calculating the annual growth.

The states of the nodes in Figure 2 are shown with equal probability distributions indicating that the BBN needs CPTs and training data to be fully functional. The belief networks are implemented using the Netica toolkit [41].

**Table 2.** States of the key indicators for the period from 1995 to 2019. The asterisks mean missing data.

| | Imports | Storage | Elec. Gen. Diversity | Energy Intensity | CO₂/Capita | Access to Elec. | Elec. Prices | Gas Prices |
|---|---|---|---|---|---|---|---|---|
| 1 | Stable | Declining | Stable | Declining | Declining | Stable | Declining | * |
| 2 | Stable | Improving | Stable | Declining | Declining | Stable | Stable | * |
| 3 | Stable | Improving | Stable | Declining | Declining | Stable | Stable | * |
| 4 | Stable | Improving | Stable | Declining | Declining | Stable | Stable | * |
| 5 | Stable | Improving | Stable | Improving | Declining | Stable | Stable | * |
| 6 | Stable | Improving | Stable | Improving | Declining | Stable | Stable | Declining |
| 7 | Stable | Improving | Stable | Improving | Declining | Stable | Stable | Stable |
| 8 | Stable | Improving | Stable | Improving | Declining | Stable | Stable | Stable |
| 9 | Stable | Improving | Stable | Improving | Declining | Stable | Stable | Stable |
| 10 | Stable | Improving | Stable | Improving | Declining | Stable | Stable | Stable |
| 11 | Stable | Improving | Stable | Improving | Declining | Stable | Stable | Stable |
| 12 | Stable | Improving | Stable | Improving | Declining | Stable | Stable | Improving |
| 13 | Stable | Improving | Stable | Improving | Declining | Stable | Stable | Stable |
| 14 | Stable | Improving | Stable | Improving | Declining | Stable | Stable | Stable |
| 15 | Stable | Improving | Stable | Improving | Declining | Stable | Stable | Stable |
| 16 | Stable | Stable | Stable | Improving | Declining | Stable | Stable | Stable |
| 17 | Stable | Stable | Stable | Improving | Declining | Stable | Stable | Stable |
| 18 | Stable | Stable | Stable | Improving | Declining | Stable | Stable | Stable |
| 19 | Stable | Improving | Stable | Improving | Declining | Stable | Stable | Stable |
| 20 | Stable | Improving | Stable | Improving | Declining | Stable | Stable | Stable |
| 21 | Stable | Improving | Stable | Declining | Declining | Stable | Stable | Stable |
| 22 | Stable | Improving | Stable | Declining | Declining | Stable | Stable | Declining |
| 23 | Stable | Improving | Stable | * | Declining | Stable | Stable | Declining |
| 24 | Stable | Improving | Stable | * | Improving | Stable | Declining | Declining |
| 25 | Stable | Stable | Improving | * | * | Stable | Declining | Declining |

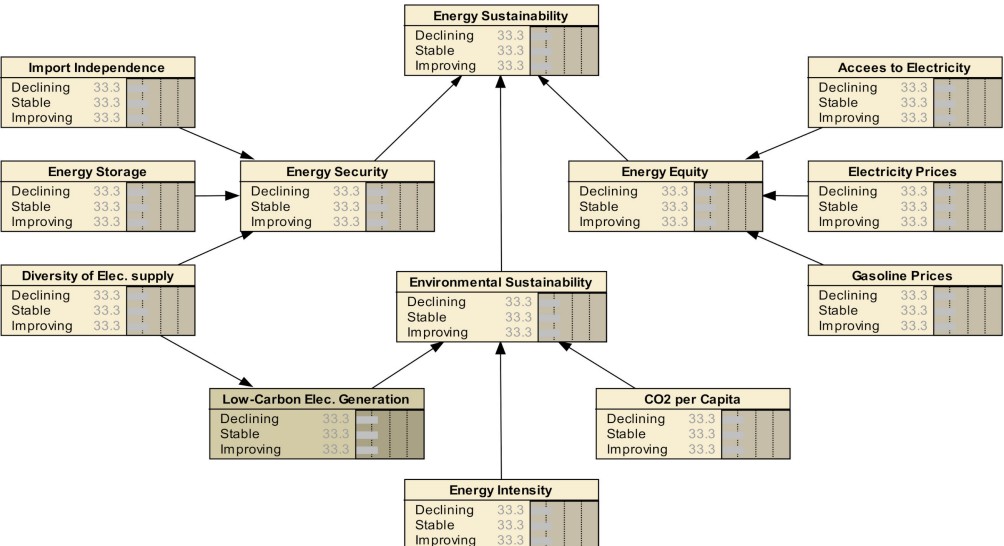

**Figure 2.** Uncompiled BBN of energy sustainability.

The CPTs for energy security, energy equity, environmental sustainability, and overall energy sustainability were created based on the weights of the variables given in the description of the WETI in Annex A of [8]. For example, the WETI gives the energy security dimension a weight of 30% distributed equally between five indicators, 6% each. Therefore, in CPT of energy security's three indicators used in this paper, equal probabilities of 0.33 were given to the declining, stable and improving statuses, which means, for instance, if the import independence and energy storage are declining but electricity generation diversity is improving, there will be 0.66 declining, 0.33 improving, and 0% stable probabilities.

For the effect variables, the size of the CPTs is the multiplication product of the numbers of the states of the effect and all its cause nodes. For the causes, it can be described by a marginal probability distribution. The probability tables are given in Appendix A.

The available data for the new policies for the period from 2018 to 2037 was mainly obtained from the energy policy simulator [42] jointly developed by Energy Innovation Policy and Technology LLC and King Abdullah Petroleum Studies and Research Center. The energy policy simulator presents data till 2050, nevertheless, the data of 20 cases (2018–2037) were used to provide a statistically acceptable representation for the near future period. The evaluation of the data to draw the states of the variables was done case by case. For example, the obtained electricity prices were for 2020, 2030, and 2035, so the periods between each interval were split between two states to describe the gradual increase or decrease.

The interconnections between the indicators were considered to recoup the missing data. The energy intensity and the $CO_2$ emission per capita were calculated based on energy efficiency, energy equity, and low-carbon electricity generation. The information of the interconnections is based on an analysis of Saudi Arabia's $CO_2$ emissions drop in 2018 [43]. The energy efficiency was given improving states until 2030 and then stable states to 2037 based on the information in the Saudi Energy Efficiency Program that efficiency will be improved to reach a 20% consumption reduction by 2030 [44].

The states of the indicators are given in Table 3. The CPTs for energy intensity and $CO_2$ per capita are given in Appendix B.

**Table 3.** States of the key indicators for the period from 2018 to 2037.

|  | Imports | Storage | Elec. Gen. Diversity | Energy Efficiency | Access to Elec. | Elec. Prices | Gas Prices |
|---|---|---|---|---|---|---|---|
| 1 | Stable | Improving | Improving | Improving | Stable | Declining | Declining |
| 2 | Stable | Declining | Improving | Improving | Stable | Declining | Declining |
| 3 | Improving | Improving | Stable | Improving | Stable | Declining | Stable |
| 4 | Improving | Improving | Improving | Improving | Stable | Declining | Stable |
| 5 | Improving | Improving | Improving | Improving | Stable | Declining | Stable |
| 6 | Improving | Improving | Improving | Improving | Stable | Declining | Stable |
| 7 | Improving | Improving | Improving | Improving | Stable | Declining | Stable |
| 8 | Improving | Improving | Improving | Improving | Stable | Declining | Stable |
| 9 | Improving | Improving | Improving | Improving | Stable | Stable | Stable |
| 10 | Improving | Improving | Improving | Improving | Stable | Stable | Stable |
| 11 | Declining | Improving | Improving | Improving | Stable | Stable | Stable |
| 12 | Declining | Improving | Improving | Improving | Stable | Stable | Stable |
| 13 | Declining | Stable | Stable | Improving | Stable | Stable | Stable |
| 14 | Declining | Stable | Stable | Stable | Stable | Stable | Stable |
| 15 | Declining | Stable | Stable | Stable | Stable | Stable | Stable |
| 16 | Declining | Stable | Stable | Stable | Stable | Improving | Stable |
| 17 | Stable | Stable | Stable | Stable | Stable | Improving | Stable |
| 18 | Stable | Stable | Stable | Stable | Stable | Improving | Stable |
| 19 | Stable | Stable | Stable | Stable | Stable | Improving | Stable |
| 20 | Stable | Stable | Stable | Stable | Stable | Improving | Stable |

## 3. Results

Figure 3 shows the results of compiling the BBN using the 25 cases from Table 2, which reveal that the likelihood of improvement in energy sustainability was 25.5%, which is comparable to the declining likelihood of 23.8%, while the most likely prospect was the stability of the existing situation with a 50.6% chance.

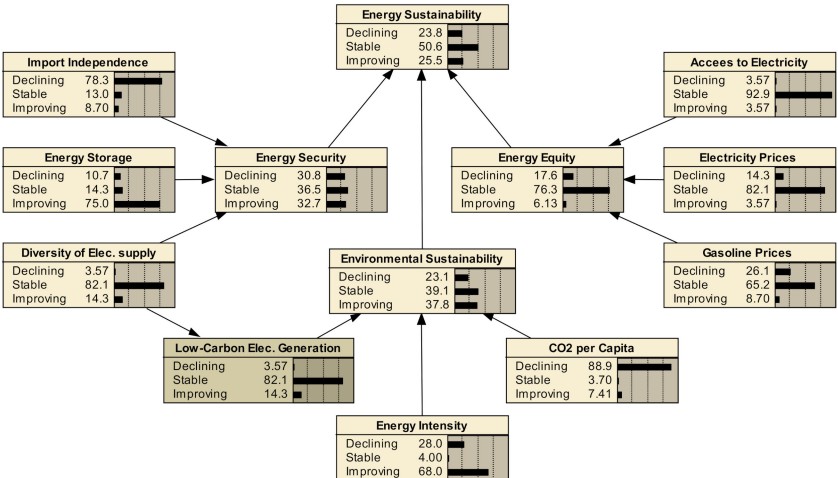

**Figure 3.** Compiled energy sustainability network for the period from 1995 to 2019.

Netica was used to carry out a sensitivity analysis, which revealed that the dependence of energy sustainability on energy security is comparable to that on environmental sustainability, and the strength of the energy equity effect is half of that of the other two dimensions. By taking the analysis to the level of the indicators, the most influential indicators were the diversity of the electricity supply followed by the energy intensity and then energy storage.

The used toolkit allows examining different scenarios by altering the states of the different variables. For example, a back-casting scenario was created by setting the improvement probability for energy sustainability to 100% and looking at changes imposed in the probabilities of the states of the cause variables (Figure 4). The back-casting results reaffirmed the previous sensitivity analysis. They showed that the most required improvement should be by further 12.5% in the diversity of electricity generation, which tacitly drives another 12.5% improvement in the share of low-carbon electricity generation, then 12.8% in energy intensity, and 8.2% in energy storage.

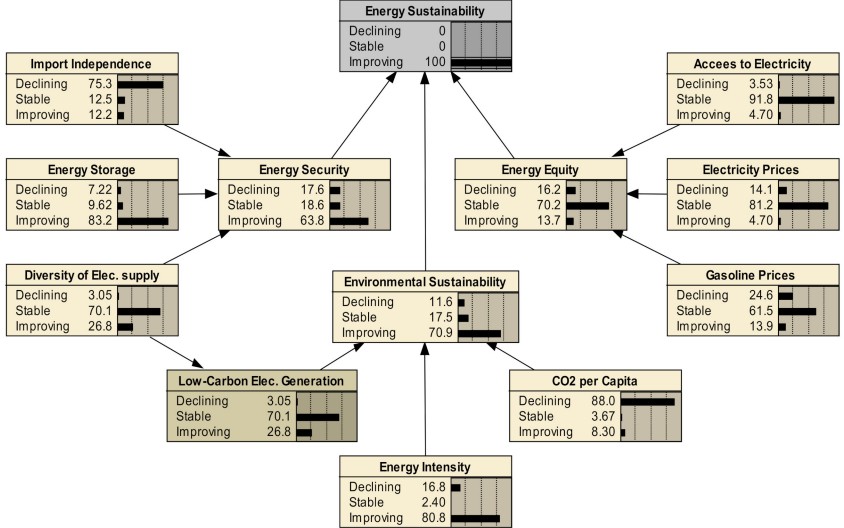

**Figure 4.** Back-casting 100% improvement probability of energy sustainability network for the period from 1995 to 2019.

Similarly, many small incremental changes in the states of any of the variables can be attempted to examine the most probable routes for the improvement of each dimension or that of the overall energy sustainability.

The compiled BBN for the BAU path did not account for the interconnections between the variables. For example, the impact of low-carbon electricity generation on the $CO_2$ emissions, fuel prices on energy intensity, and electricity generation mix on the affordability. The reason is that the paper studies the specific case of Saudi Arabia's performance, where actual data that already represent the sum measurements are available and do not need further calculation or elicitation.

However, some of the interconnections were estimated to assess the energy sustainability landscape in the light of policy changes in the Saudi 2030 Vision (Figure 5). The improvement likelihood of energy sustainability was 33.6%, with a 53% probability that the performance will be steady during the specified period. The back-casting (Figure 6) and sensitivity analysis showed that the most influential group of indicators is that composed of the diversity of electricity generation, $CO_2$ emission per capita, and energy intensity, respectively, in terms of the magnitudes of their strength. The groups in the second order of influence were energy storage and import independence with comparable strengths.

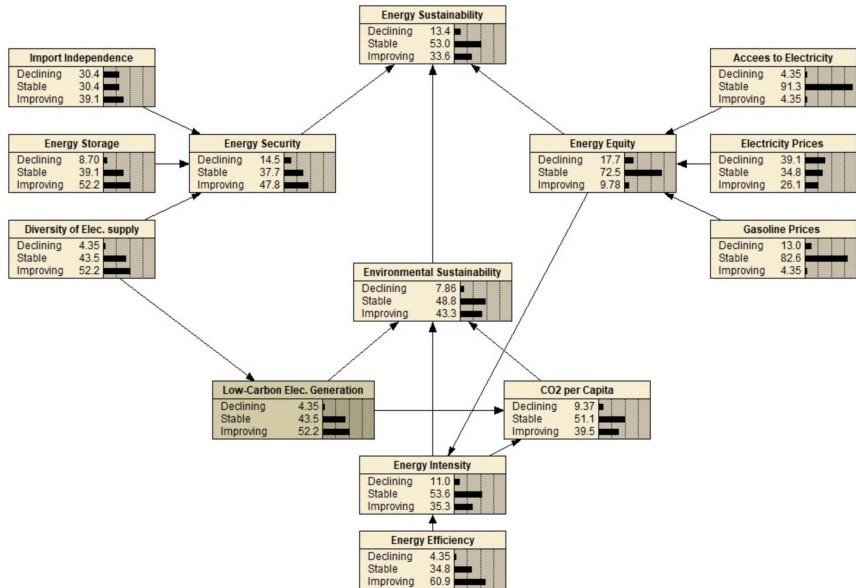

**Figure 5.** Compiled energy sustainability network for the period from 2018 to 2037.

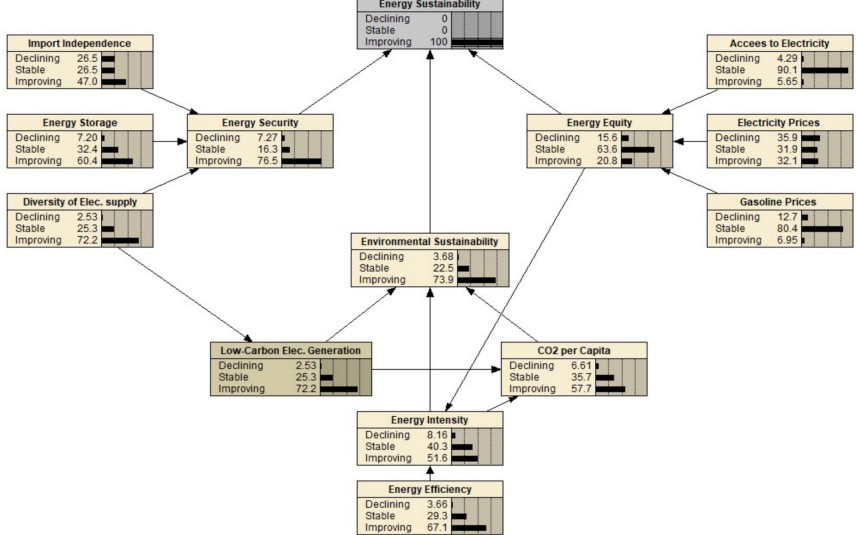

**Figure 6.** Back-casting 100% improvement probability of energy sustainability network for the period from 2019 to 2030.

## 4. Discussion

System thinking is an appropriate approach to study energy sustainability policies. Probabilistic and mathematical modeling enables a formal realization of energy sustainability dynamics.

Applying Bayesian Belief Networks to Saudi Arabia's context quantified the probability of improvement, decline, and steadiness of the Business as Usual scenario pertaining to the identified energy sustainability dimensions. A similar method was used by Daim et al. to support policy design in the state of Oregon, USA. It differs from the method in this paper in that the BBNs have been constructed by conversion from Causal Maps, which were developed by consulting experts and energy authorities in Oregon. The step of generating the Causal Maps is analogous to the adoption of the WETI indicators (causes) and dimensions (effects) in this work. The results offered networks with probabilities like those shown in Section 3, Figures 3–6, describing the different states of the correlated factors [29].

The BBN application presented in [28] has sought a higher accuracy in the construction of the BBN by applying an augmented naive model to the quantitative data and k-folds analysis of the Bayesian models. The researchers examined the best scenarios to inform policymaking in Italy and Germany concerning geothermal energy and hydro energy. Another study aimed at computing probabilities of power system states to enable renewable energy integration into smart grids and improve power flow control. The approach addressed a more sophisticated issue of real-time modeling of power systems [31].

The scope of the previous studies was limited to exploring the scenarios of nuclear energy, renewable energy, and investments in different contexts. The contribution of this work is the use of BBN to model the entire energy sustainability system covering energy security, energy equity, and environmental sustainability.

This research was performed before the release of 2020 WETI. Saudi Arabia's 2020 profile shows slight changes compared to 2019 scores: From 55 to 59.9 in energy security, from 98 to 99 in energy equity, and from 35 to 44.3 in environmental sustainability [45]. The change extent supports the findings of the research depicted in Figure 3 and described in Section 3 concerning the energy policy Business as Usual scenario.

For the 2030 policies, the study provided a tool to devise numerous states of each policy and evaluate the propagation of their impact to assist in identifying the critical engagements to achieve energy sustainability. For instance, $CO_2$ per capita and energy intensity are interlinked with the parent variables being energy efficiency and energy equity, and the impact of the latter was mainly due to the fluctuations in the electricity prices. Therefore, the joint impact of energy efficiency and energy prices reform needs equal consideration to that given to energy resources diversification.

BBN is a useful decision support tool that can give insights and an improved understanding of the policy context and allows the examination of several alternatives. The sensitivity analysis that measures the interdependencies between the BBN nodes gives specific information that the policymakers can use to plan the desired adjustments for improved sustainability.

For enhanced accuracy in analyzing the new energy policies, a more complex BBN comprising the 32 indicators of the WETI and their causes is suggested. The most influential indicators identified in the suggested complex BBN can be further investigated to disclose the required finer interventions. Moreover, other system dynamic methods can be applied to attain different perceptions of interdependence.

The projected data can also be reinforced by experts' judgment and stakeholder consultations to assist the policymakers in optimizing the options. This type of qualitative data can be used in BBN following methods like those described in [16,46] (pp. 11–12).

**Author Contributions:** Conceptualization, M.S.H.M., A.A. and A.Y.S.; sata curation, M.S.H.M.; investigation, M.S.H.M.; methodology, M.S.H.M.; project administration, A.A.; supervision, M.S.H.M.; validation, M.S.H.M.; writing—original draft, M.S.H.M.; writing—review & editing, M.S.H.M. and A.A. All authors have read and agreed to the published version of the manuscript.

**Funding:** This project was funded by the Deanship of Scientific Research (DSR) at King Abdulaziz University, Jeddah, under grant No. (J: 65-135-1441). The authors, therefore, acknowledge with thanks DSR for technical and financial support.

**Institutional Review Board Statement:** Not applicable.

**Informed Consent Statement:** Not applicable.

**Data Availability Statement:** Data set available on request to corresponding authors.

**Conflicts of Interest:** The authors declare no conflict of interest.

## Appendix A. Conditional Probability Tables (CPTs) for the Energy Sustainability Network for the Period from 1995 to 2019

**Table A1.** Energy security CPT.

| Import Independence | Energy Storage | Diversity of Elec. Supply | Declining | Stable | Improving |
|---|---|---|---|---|---|
| Declining | Declining | Declining | 100 | 0 | 0 |
| Declining | Declining | Stable | 66.666 | 33.334 | 0 |
| Declining | Declining | Improving | 66.666 | 0 | 33.334 |
| Declining | Stable | Declining | 66.666 | 33.334 | 0 |
| Declining | Stable | Stable | 33.334 | 66.666 | 0 |
| Declining | Stable | Improving | 33.333 | 33.333 | 33.333 |
| Declining | Improving | Declining | 66.666 | 0 | 33.334 |
| Declining | Improving | Stable | 33.333 | 33.333 | 33.333 |
| Declining | Improving | Improving | 33.333 | 0 | 66.667 |
| Stable | Declining | Declining | 66.666 | 33.334 | 0 |
| Stable | Declining | Stable | 33.334 | 66.666 | 0 |
| Stable | Declining | Improving | 33.333 | 33.333 | 33.333 |
| Stable | Stable | Declining | 33.334 | 66.666 | 0 |
| Stable | Stable | Stable | 0 | 100 | 0 |
| Stable | Stable | Improving | 0 | 66.666 | 33.334 |
| Stable | Improving | Declining | 33.333 | 33.333 | 33.333 |
| Stable | Improving | Stable | 0 | 66.666 | 33.334 |
| Stable | Improving | Improving | 0 | 33.333 | 66.667 |
| Improving | Declining | Declining | 66.666 | 0 | 33.334 |
| Improving | Declining | Stable | 33.333 | 33.333 | 33.333 |
| Improving | Declining | Improving | 33.333 | 0 | 66.667 |
| Improving | Stable | Declining | 33.333 | 33.333 | 33.333 |
| Improving | Stable | Stable | 0 | 66.666 | 33.334 |
| Improving | Stable | Improving | 0 | 33.333 | 66.667 |
| Improving | Improving | Declining | 33.333 | 0 | 66.667 |
| Improving | Improving | Stable | 0 | 33.333 | 66.667 |
| Improving | Improving | Improving | 0 | 0 | 100 |

**Table A2.** Energy equity CPT.

| Gasoline Prices | Electricity Prices | Access to Electricity | Declining | Stable | Improving |
|---|---|---|---|---|---|
| Declining | Declining | Declining | 100 | 0 | 0 |
| Declining | Declining | Stable | 75 | 25 | 0 |
| Declining | Declining | Improving | 75 | 0 | 25 |
| Declining | Stable | Declining | 75 | 25 | 0 |
| Declining | Stable | Stable | 50 | 50 | 0 |
| Declining | Stable | Improving | 50 | 25 | 25 |
| Declining | Improving | Declining | 75 | 0 | 25 |
| Declining | Improving | Stable | 50 | 25 | 25 |
| Declining | Improving | Improving | 50 | 0 | 50 |
| Stable | Declining | Declining | 50 | 50 | 0 |
| Stable | Declining | Stable | 25 | 75 | 0 |
| Stable | Declining | Improving | 50 | 25 | 25 |
| Stable | Stable | Declining | 25 | 75 | 0 |
| Stable | Stable | Stable | 0 | 100 | 0 |
| Stable | Stable | Improving | 0 | 75 | 25 |
| Stable | Improving | Declining | 25 | 50 | 25 |
| Stable | Improving | Stable | 0 | 75 | 25 |
| Stable | Improving | Improving | 0 | 50 | 50 |
| Improving | Declining | Declining | 50 | 0 | 50 |
| Improving | Declining | Stable | 25 | 25 | 50 |
| Improving | Declining | Improving | 25 | 0 | 75 |
| Improving | Stable | Declining | 25 | 25 | 50 |
| Improving | Stable | Stable | 0 | 50 | 50 |
| Improving | Stable | Improving | 0 | 25 | 75 |
| Improving | Improving | Declining | 25 | 0 | 75 |
| Improving | Improving | Stable | 0 | 25 | 75 |
| Improving | Improving | Improving | 0 | 0 | 100 |

**Table A3.** Environmental sustainability CPT.

| Low-Carbon Elec. Generation | Energy Intensity | $CO_2$ per Capita | Declining | Stable | Improving |
|---|---|---|---|---|---|
| Declining | Declining | Declining | 100 | 0 | 0 |
| Declining | Declining | Stable | 90 | 10 | 0 |
| Declining | Declining | Improving | 90 | 0 | 10 |
| Declining | Stable | Declining | 55 | 45 | 0 |
| Declining | Stable | Stable | 45 | 55 | 0 |
| Declining | Stable | Improving | 45 | 45 | 10 |
| Declining | Improving | Declining | 55 | 0 | 45 |
| Declining | Improving | Stable | 45 | 10 | 45 |
| Declining | Improving | Improving | 45 | 0 | 55 |
| Stable | Declining | Declining | 55 | 45 | 0 |
| Stable | Declining | Stable | 45 | 55 | 0 |
| Stable | Declining | Improving | 45 | 45 | 10 |
| Stable | Stable | Declining | 10 | 90 | 0 |
| Stable | Stable | Stable | 0 | 100 | 0 |
| Stable | Stable | Improving | 0 | 90 | 10 |
| Stable | Improving | Declining | 10 | 45 | 45 |
| Stable | Improving | Stable | 0 | 55 | 45 |
| Stable | Improving | Improving | 0 | 45 | 55 |
| Improving | Declining | Declining | 55 | 0 | 45 |
| Improving | Declining | Stable | 45 | 10 | 45 |
| Improving | Declining | Improving | 45 | 0 | 55 |
| Improving | Stable | Declining | 10 | 45 | 45 |
| Improving | Stable | Stable | 0 | 55 | 45 |
| Improving | Stable | Improving | 0 | 45 | 55 |
| Improving | Improving | Declining | 10 | 0 | 90 |
| Improving | Improving | Stable | 0 | 10 | 90 |
| Improving | Improving | Improving | 0 | 0 | 100 |

**Table A4.** Overall energy sustainability CPT.

| Energy Equity | Energy Security | Environmental Sustainability | Declining | Stable | Improving |
|---|---|---|---|---|---|
| Declining | Declining | Declining | 100 | 0 | 0 |
| Declining | Declining | Stable | 66.667 | 33.333 | 0 |
| Declining | Declining | Improving | 66.667 | 0 | 33.333 |
| Declining | Stable | Declining | 66.667 | 33.333 | 0 |
| Declining | Stable | Stable | 33.333 | 66.667 | 0 |
| Declining | Stable | Improving | 33.333 | 33.333 | 33.333 |
| Declining | Improving | Declining | 66.667 | 0 | 33.333 |
| Declining | Improving | Stable | 33.333 | 33.333 | 33.333 |
| Declining | Improving | Improving | 33.333 | 0 | 66.667 |
| Stable | Declining | Declining | 66.667 | 33.333 | 0 |
| Stable | Declining | Stable | 33.333 | 66.667 | 0 |
| Stable | Declining | Improving | 33.333 | 33.333 | 33.333 |
| Stable | Stable | Declining | 33.333 | 66.337 | 0 |
| Stable | Stable | Stable | 0 | 100 | 0 |
| Stable | Stable | Improving | 0 | 66.667 | 33.333 |
| Stable | Improving | Declining | 33.333 | 33.333 | 33.333 |
| Stable | Improving | Stable | 0 | 66.667 | 33.333 |
| Stable | Improving | Improving | 0 | 33.333 | 66.667 |
| Improving | Declining | Declining | 66.667 | 0 | 33.333 |
| Improving | Declining | Stable | 33.333 | 33.333 | 33.333 |
| Improving | Declining | Improving | 33.333 | 0 | 66.667 |
| Improving | Stable | Declining | 33.333 | 33.333 | 33.333 |
| Improving | Stable | Stable | 0 | 66.667 | 33.333 |
| Improving | Stable | Improving | 0 | 33.333 | 66.667 |
| Improving | Improving | Declining | 33.333 | 0 | 66.667 |
| Improving | Improving | Stable | 0 | 33.333 | 66.667 |
| Improving | Improving | Improving | 0 | 0 | 100 |

## Appendix B. CPTs for the Energy Intensity and $CO_2$ per Capita for the Period from 2018 to 2037

**Table A5.** Energy intensity CPT.

| Energy Equity | Energy Efficiency | Declining | Stable | Improving |
|---|---|---|---|---|
| Declining | Declining | 100 | 0 | 0 |
| Declining | Stable | 50 | 50 | 0 |
| Declining | Improving | 50 | 0 | 50 |
| Stable | Declining | 50 | 50 | 0 |
| Stable | Stable | 0 | 100 | 0 |
| Stable | Improving | 0 | 50 | 50 |
| Improving | Declining | 50 | 0 | 50 |
| Improving | Stable | 0 | 50 | 50 |
| Improving | Improving | 0 | 0 | 100 |

**Table A6.** $CO_2$ emissions per capita CPT.

| Energy Equity | Energy Efficiency | Declining | Stable | Improving |
|---|---|---|---|---|
| Declining | Declining | 100 | 0 | 0 |
| Declining | Stable | 75 | 25 | 0 |
| Declining | Improving | 75 | 0 | 25 |
| Stable | Declining | 25 | 75 | 0 |
| Stable | Stable | 0 | 100 | 0 |
| Stable | Improving | 0 | 75 | 25 |
| Improving | Declining | 25 | 0 | 75 |
| Improving | Stable | 0 | 25 | 75 |
| Improving | Improving | 0 | 0 | 100 |

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
