# Peer review of "An Integrated Approach to the Realization of Saudi Arabia’s Energy Sustainability"

_sustainability, doi:10.3390/su13010205_

Round 1

Reviewer 1 Report

The work does not provide sufficient references in the field of energy sustainability analysis. Several works have been done and different method proposed. Those works should be cited, referenced and used as comparison tool to understand the significance of the present article and related work.

This lack of appropriate background and specific reference analysis reflects in not-yet-acceptable results presentation, discussion and overall significance of the results.

Author Response

Thank you for the constructive comment.

As a response, eight new references have been cited and briefly discussed. Please see the Introduction Section lines 39-57 and 66-75.

Reviewer 2 Report

  1. This paper introduces an integrated energy sustainability index system for Saudi Arabia. As Figures 2-6 show, there are many subdirectories with numbers to watch.  This system is informative and useful to trace and watch the energy sustainability for a country.  Some minor problems remain to clarify or improve.
  2. As Eq. (1) shows, the most important mathematical foundation for this system is Bayesian probability. Is there any numerical example show how the future values or scores are obtained through applying the Bayesian probability?
  3. Figures 3-6 report many predictions of the values or scores to the year 2030. There should be a system of equations behind this integrated system.  The authors should display some most important equations to show how this whole system works to generate predictions about the future.
  4. Again, the Appendix reports many declining and improving trends for different subdirectories. It is still not clear how these predictions were generated.  A summarized list of key equations and also important parameter values is needed to justify the generation of these trend predictions.
  5. As Figures 2-6 show, there are many subdirectories with scores or numbers. Is it possible to give weights to these subdirectories such that an aggregate score can be obtained as the energy sustainability index score for Saudi Arabia?
  6. As Table 3 and Figures 3-6 show, there are many improving and declining indexes for Saudi Arabia till the year of 2030. Is there any focus that the energy policy of Saudi Arabia should pay attention to?
  7. Is there any similar integrated energy sustainability system in the world? These similar energy sustainability systems should be briefly cited and compared.
  8. Most of the references are data sources and technical reports. More academic journal articles on energy sustainability index system should be cited and briefly introduced.

Author Response

  • This paper introduces an integrated energy sustainability index system for Saudi Arabia. As Figures 2-6 show, there are many subdirectories with numbers to watch.  This system is informative and useful to trace and watch the energy sustainability for a country.  Some minor problems remain to clarify or improve.

Thank you for the encouraging comment.

  • Again, the Appendix reports many declining and improving trends for different subdirectories. It is still not clear how these predictions were generated.  A summarized list of key equations and also important parameter values is needed to justify the generation of these trend predictions.

The Conditional Probability Tables were not generated using equations. For a better clarification, an explanatory text has been added (lines 180-188) about the method adopted to generate the tables.

  • Is there any similar integrated energy sustainability system in the world? These similar energy sustainability systems should be briefly cited and compared.

Eight new references have been cited and briefly discussed. Please see the Introduction Section lines 39-57 and 66-75

  • Most of the references are data sources and technical reports. More academic journal articles on energy sustainability index system should be cited and briefly introduced.

The new eight reference we added are all research journals articles

  • As Table 3 and Figures 3-6 show, there are many improving and declining indexes for Saudi Arabia till the year of 2030. Is there any focus that the energy policy of Saudi Arabia should pay attention to?

The energy policy focus has already been mentioned in the 3rd paragraph of the Discussion Section. Please see lines 272-273 of the manuscript “Therefore, the joint impact of energy efficiency and energy prices reform needs equal consideration to that given to energy resources diversification.”

Figures 3-6 report many predictions of the values or scores to the year 2030. There should be a system of equations behind this integrated system.  The authors should display some most important equations to show how this whole system works to generate predictions about the future.

The results are not predictions themselves; they are probabilities that are calculated based on predicted data. The predicted or projected data was obtained from other sources that are cited in the manuscript.

  • As Eq. (1) shows, the most important mathematical foundation for this system is Bayesian probability. Is there any numerical example show how the future values or scores are obtained through applying the Bayesian probability?

Bayes' theorem is not used to obtain future values, it is used to calculate the conditional probabilities, then the joint probabilities are calculated to give the belief.

As Figures 2-6 show, there are many subdirectories with scores or numbers. Is it possible to give weights to these subdirectories such that an aggregate score can be obtained as the energy sustainability index score for Saudi Arabia?

Like we stressed in the manuscript, the proposed analysis is of probabilistic nature, so; the results are given as a probability of change in energy sustainability state rather than a deterministic score of a state.

Round 2

Reviewer 1 Report

The paper includes in the revised version some references and a short and superficial review of some previous works and approaches to assess energy sustainability.

The new content is added to the previous content and not used to better and more deeply compare pros and cons of the proposed method with the other indicated approaches.

It is not good enough yet in terms of general significance and usefulness of the content.

Author Response

Dear Sir/Madam,

For the sustainability analysis, we briefly reviewed references 3-6 in Revision 1 of the paper. The brief review is because the references, although claiming to address energy sustainability, their contents addressed limited elements of the sustainability system, unlike the methodology of the WETI that we adopted in the paper.

The objective of the paper is to model a recognized sustainability analysis approach and use the results to comprehend a specific context, rather than proposing a new method.

To our knowledge, three publications discussed the adopted WETI, and they have been reviewed in Revision1. The previous review has been edited; please see the red-coloured text in the Introduction Section.

Thanks to your comment, we realized that a review of the BBN energy-related applications in previous studies should be added. Again, only a few references could be located in the literature. Please see the red-coloured text in the Discussion Section.

Kind regards.

Round 3

Reviewer 1 Report

The paper can be accepted for publication.